# *Plasmodium falciparum* CLAG Paralogs All Traffic to the Host Membrane but Knockouts Have Distinct Phenotypes

**DOI:** 10.3390/microorganisms12061172

**Published:** 2024-06-08

**Authors:** Ankit Gupta, Zabdi Gonzalez-Chavez, Sanjay A. Desai

**Affiliations:** Laboratory of Malaria and Vector Research, National Institute of Allergy and Infectious Diseases, National Institutes of Health, Rockville, MA 20852, USA

**Keywords:** *Plasmodium falciparum*, malaria, multigene family, CLAG, nutrient uptake, ion channel, DNA transfection

## Abstract

Malaria parasites increase their host erythrocyte’s permeability to obtain essential nutrients from plasma and facilitate intracellular growth. In the human *Plasmodium falciparum* pathogen, this increase is mediated by the plasmodial surface anion channel (PSAC) and has been linked to CLAG3, a protein integral to the host erythrocyte membrane and encoded by a member of the conserved *clag* multigene family. Whether paralogs encoded by other *clag* genes also insert at the host membrane is unknown; their contributions to PSAC formation and other roles served are also unexplored. Here, we generated transfectant lines carrying epitope-tagged versions of each CLAG. Each paralog is colocalized with CLAG3, with concordant trafficking via merozoite rhoptries to the host erythrocyte membrane of newly invaded erythrocytes. Each also exists within infected cells in at least two forms: an alkaline-extractable soluble form and a form integral to the host membrane. Like CLAG3, CLAG2 has a variant region cleaved by extracellular proteases, but CLAG8 and CLAG9 are protease resistant. Paralog knockout lines, generated through CRISPR/Cas9 transfection, exhibited uncompromised growth in PGIM, a modified medium with higher physiological nutrient levels; this finding is in marked contrast to a recently reported CLAG3 knockout parasite. CLAG2 and CLAG8 knockout lines exhibited compensatory increases in the transcription of the remaining *clags* and associated *rhoph* genes, yielding increased PSAC-mediated uptake for specific solutes. We also report on the distinct transport properties of these knockout lines. Similar membrane topologies at the host membrane are consistent with each CLAG paralog contributing to PSAC, but other roles require further examination.

## 1. Introduction

*Plasmodium* spp. are successful pathogens of humans and other vertebrates. These obligate intracellular parasites have complex life cycles and cause malaria through invasion and replication within host erythrocytes. Most species export a range of proteins to remodel their host cells to allow cytoadherence, immune evasion, and nutrient uptake [1,2,3]. These exported proteins are often encoded by multigene families that facilitate antigenic variation and evasion of host immunity [4,5,6]. Notably, members of these families are typically in subtelomeric regions, where active recombination produces new variants that increase diversity [7,8]. Most of these multigene families encode proteins that function in cytoadherence or immune evasion [9]. Interestingly, most are also restricted to one or a small number of species. For example, the extensively studied *var* genes in the *P. falciparum* parasite are restricted to this human pathogen, and a small number of species infecting African primates; *rif, vir*, *yir,* and *sicavar* gene families are also each present in a limited number of *Plasmodium* spp. [10,11].

An important exception with a distinct function is the *clag* multigene family (cytoadherence linked asexual gene), which is conserved in all examined members of the *Plasmodium* genus. While the 3D7 reference line of *P. falciparum* has 5 copies, other species have between 2 and over 35 copies of the *clag* genes [12,13]. Most *P. falciparum* clones carry two copies on chromosome 3 (termed *clag3.1* and *clag3.2*), but some lines have undergone copy number reduction to yield a single hybrid *clag3h* [14]. Interestingly, in vitro selection has also yielded a line with three *clag3* genes [15]. Three other copies in the 3D7 line (*clag2*, *clag8*, and *clag9*) are at subtelomeric sites on chromosomes 2, 8, and 9, respectively. Some lines carry additional *clag* gene copies [12], indicating an ongoing evolution of this conserved gene family. Adding to these complexities, *clag* genes undergo complicated epigenetic regulation with monoallelic expression and switching of *clag3* paralogs; *clag2*, but not *clag8* and *clag9*, has also been clearly shown to have a variable expression in *P. falciparum* clones [16].

The encoded CLAG proteins were originally proposed to function in cytoadherence. based on molecular and biochemical studies of a chromosome 9 deletion event associated with loss of infected cell binding to the CD36 receptor [17]. While some subsequent studies supported a role for CLAG9 in cytoadherence [18], this observation was cast into doubt [19]. Transcription in schizonts and packaging into rhoptries also has led to suggested roles in erythrocyte invasion and/or formation of the parasitophorous vacuole that surrounds the intracellular parasite [20], but direct experimental evidence supporting these roles is missing.

In surprising contrast to these reports, more recent studies have instead strongly implicated a CLAG3 role in the formation of the plasmodial surface anion channel (PSAC), a nutrient uptake channel on the host membrane [21]. A role in PSAC-mediated nutrient uptake was first identified by unbiased genetic mapping studies with ISPA-28, a small molecule inhibitor that selectively blocks channels associated with the *clag3.1* gene product from the Dd2 parasite line but is inactive against channels associated with the Dd2 *clag3.2* or either *clag3* paralog in other parasites [22]. Additional genetic mapping studies using either the channel’s protease susceptibility or growth inhibition in PGIM, a nutrient-restricted medium, also mapped the two *clag3* genes [15,23]. Characterization of transport mutants generated by in vitro selection has revealed mutations in *clag3* or epigenetic silencing of *clag3* [24,25], providing independent evidence for a role in nutrient uptake.

By comparison, much less is known about the other *clag* genes in *P. falciparum*. Indeed, the failure of genetic mapping and transport mutants to implicate these other paralogs could suggest that they serve unrelated roles in blood-stage parasites. Here, we used molecular and biochemical studies to examine the trafficking, localization, and possible roles of the proteins encoded by *clag2*, *clag8*, and *clag9*. Our studies reveal that each CLAG traffics to rhoptries and is transferred to the next erythrocyte at invasion. Each reaches the host membrane with peripheral and integral membrane pools, as previously determined for CLAG3. Our protease susceptibility studies reveal that CLAG2, but not CLAG8 or CLAG9, is susceptible to extracellular protease, consistent with a larger variant domain amongst *P. falciparum* lines. Most importantly, our molecular and biochemical studies provide experimental evidence supporting the contributions of CLAG2 and CLAG8 in PSAC formation. As with CLAG3, conclusive evidence for the direct contribution to the formation of the channel pore remains elusive; other roles could also not be excluded by our studies and should, therefore, continue to be explored. These findings provide a novel example of a conserved gene family that serves essential roles in intracellular parasite development. Our studies also provide insights into PSAC pharmacology that should guide antimalarial drug development against this essential channel.

## 2. Materials and Methods

### 2.1. Parasite Culture

*P. falciparum* laboratory lines were cultivated at 37 °C under 5% O_2_, 5% CO_2_, and 90% N_2_ in O^+^ human erythrocytes (Interstate Blood Bank) using RPMI-1640 supplemented with 25 mM HEPES, 50 μg/mL hypoxanthine (KD Medical), 0.5% NZ Microbiological BSA (MP Biomedicals), and 0.23% NaHCO_3_ (Gibco). The modified medium, with lower, more physiological concentrations of isoleucine (11.4 µM) and hypoxanthine (3.01 µM), was similarly used and prepared as described previously [15].]

### 2.2. Production of Engineered Transfectant Lines

CLAG paralog C-terminal 3xFLAG epitope-tagged lines were produced by CRISPR/Cas9 transfection using *pL6-hdhfr* plasmids. InFusion cloning (Clontech, Mountain View, CA, USA) was used to introduce high-scoring sgRNA for expression under the *Pf*U6 promoter, as listed in Appendix A. A double-stranded DNA construct carrying 5′ and 3′ homology arms of 250–500 bp length each, shield mutations at the protospacer site targeted by Cas9, a 3x FLAG epitope tag, and an in-frame stop codon was synthesized and introduced by InFusion cloning. Cas9 was expressed from a separate *pUF1-Cas9* plasmid. After transfection of *C3-TetR* by standard methods [26], cultures were selected with 1.5 nM WR99210 and 1.5 µM DSM-1 [27,28]. All reported experiments were performed using limiting dilution clones that were confirmed by PCR and DNA sequencing.

We also produced CLAG paralog knockout lines using CRISPR/Cas9 transfection with high-scoring sgRNA near the gene 5′ end, as listed in Appendix A. Here, synthetic DNA constructs disrupted the ORF near the gene’s 5′ end by introducing stop codons and/or internal deletions of the gene. The *hdhfr* cassette was inserted between the two homology arms in the case of the *clag2* gene knockout. These knockouts were produced in the wild-type KC5 line and cloned as described above for the tagged lines.

### 2.3. Indirect Immunofluorescence Microscopy

Confocal immunofluorescence microscopy images were obtained using thin smears of parasite cultures fixed in chilled 1:1 acetone–methanol for 5 min. After fixation, slides were blocked with 3% skim milk in PBS for 1 h at RT. Slides were probed with mouse M2 mouse anti-FLAG monoclonal antibody (Sigma Aldrich, St. Louis, MO, USA) at 1:250 dilution and rabbit anti-HA antibody (Abcam, Cambridge, UK) at 1:100 dilution for 1 h at RT or overnight at 4 °C. After washing with cold PBS, slides were incubated with 2 µg/µL DAPI (4′, 6-diamidino-2-phenylindole), goat anti-mouse AF488 and goat anti-rabbit AP594 at 1:500 dilution for 30 min at RT, washed with ice-cold PBS and mounted with Prolong Diamond anti-fade mountant (Molecular Probes, Eugene, OR, USA). Images were collected using a 64× oil immersion objective on a Leica SP5 or SP8 confocal microscope and processed using Leica LAS X software (https://www.leica-microsystems.com/; accessed 15 December 2016).

### 2.4. Immunoblots

Trophozoite-stage cultures were percoll-sorbitol enriched to ≥96% parasitemia and used for immunoblotting. Where used, protease susceptibility was performed using washed cells at 5% hematocrit and 1 mg/mL pronase E from *Streptomyces griseus* (Sigma Aldrich) in PBS supplemented with 0.6 mM CaCl_2_ and 1 mM MgCl_2_ for 1 h at 37 °C. Reactions were terminated by adding ice-cold PBS2 with 2 mM PMSF and 2 mM EGTA, followed by extensive washing. Cells were then lysed in ice-cold hypotonic lysis buffer (7.5 mM NaHPO_4_, 1 mM EDTA, pH 7.5) with 1 mM PMSF and ultra-centrifugation (100,000× *g*, 4 °C, 1 h). The supernatant was collected as the soluble fraction and the remaining pellet was resuspended and incubated in 100 mM Na_2_CO_3_ at pH 11 for 30 min at 4 °C. After ultracentrifugation, the supernatant was collected as the carbonate extract and the pellet as the integral membrane fraction. After neutralization with 1/10 volume of 1 M HCl, SDS loading buffer was added prior to protein separation by SDS-PAGE (4 to 15% Mini-Protean TGX gel, Bio-Rad, Hercules, CA, USA) and transferred to nitrocellulose membrane. Membranes were blocked and incubated with anti-CLAG3, 1:2000 dilution; M2 anti-FLAG, 1:8000 dilution, or HRP-conjugated anti-plasmodium aldolase (Abcam), 1:2500 dilution. After washing, horseradish peroxidate (HRP)-conjugated secondary antibody was applied (anti-mouse IgG, 1:10,000 dilution, Sigma Aldrich) with Clarity Western ECL substrate (Bio-Rad). Binding was detected on Hyblot X-ray film. Band intensities were measured from independent trials using ImageJ software (https://imagej.nih.gov/, 15 December 2016) and normalized to aldolase loading control band intensities.

### 2.5. Growth Inhibition Assay

Parasite growth rates in standard RPMI 1640-based medium and PGIM were quantified using SYBR Green I, as described previously [29]. Synchronized ring-stage cultures at 0.2% parasitemia and 2.5% hematocrit were cultivated in these media for 5 days, with a medium change after 2 days. At harvest, cultures were lysed in 20 mM Tris pH 7.5, 10 mM EDTA, 1.6% Triton X-100, and 0.016% saponin at pH 7.5 with a 2500-fold dilution of SYBR Green I nucleic acid gel stain (ThermoFisher, Waltham, MA, USA). After a 30 min incubation in the dark, fluorescence was measured at 485 nm excitation/528 nm emission to quantify parasite nucleic acid production (BioTex Synergy HT microplate reader, Santa Clara, CA, USA). Expansion of parasite cultures in triplicate wells was normalized to matched cultures killed with 20 µM chloroquine.

### 2.6. Quantitative Real-Time PCR

We used quantitative real-time PCR (qRT-PCR) to measure changes in the expression of *rhoph* genes in *clag* knockout lines. Total RNA was harvested from synchronized late trophozoite-stage infected cells 27 h after sorbitol synchronization. Homogeneity and matching of parasite stages were confirmed by microscopic examination of smears for all lines. We have validated this synchronization protocol using *msp2*, an unrelated gene whose stage-specific transcription matches that of the *clag*, *rhoph2*, and *rhoph3* genes [29].

The qRT-PCR was performed as described previously [29] using primers listed in Appendix A, prepared cDNA, and the QuantiTect SyBR Green kit (Qiagen, Hilden, Germany). Primers for qRT-PCR were designed based on specificity for each *rhoph* gene and a desired amplicon size of ~120 bp. After DNase I treatment to remove genomic DNA (TURBO DNA-free kit, Ambion, Seoul, Republic of Korea), first-strand cDNA was synthesized using ~1.5 µg RNA, oligo-dT primers, and Superscript III reverse transcriptase (Invitrogen, Waltham, MA, USA). Diluted cDNA was used for qRT-PCR in the iCycler IQ multicolor real-time PCR system (Bio-Rad) using a three-step program: denaturation at 95 °C for 15 min followed by 40 cycles of annealing at 52 °C and extension at 62 °C for 30 s each. The final stage used gradual heating from 55 to 95 °C, incrementing by 0.5 °C for 30 s; this dissociation protocol was used to confirm the specificity of primer binding and product synthesis. Each qRT-PCR reaction was accompanied by a negative control (-RT) to exclude gDNA contamination. All reactions were performed in triplicate with the average threshold cycle (C_T_) values from 4 independent RNA harvests from each knockout clone and the wild-type parent. The constitutively expressed Pf07_0073 gene was used as a loading control. Normalized gene expression was estimated according to 2^(mean C_T_ value of the *rhoph* gene − mean C_T_ value of Pf07_0073) and is presented as the mean ± S.E.M. from independent harvests. Significance was evaluated using ordinary one-way ANOVA testing with Dunnett’s multiple comparisons test (GraphPad Prism 9.4).

### 2.7. Osmotic Lysis Transport Assays

PSAC-mediated transport of organic solutes and the effects of inhibitors or protease treatment were tracked using continuous tracking of 700 nm light transmittance through cell suspensions, as described previously [22]. Light transmittance through the suspension increases with infected cell osmotic lysis due to channel-mediated solute uptake. Percoll-sorbitol enriched trophozoite-stage infected cells were washed and resuspended in 140 mM NaCl, 20 mM Na-HEPES, and 0.1 mg/mL bovine serum albumin at pH 7.4. Net solute uptake and osmotic lysis were initiated by the addition of 20 volumes of 20 mM Na-HEPES and 0.1 mg/mL bovine serum albumin at pH 7.4 with 280 mM sorbitol, 280 mM proline, or 145 mM PhTMA^+^Cl^−^, solutes with high PSAC permeability. Pretreatment with pronase E to evaluate channel protease susceptibility was performed as described above; PSAC inhibitors were added from DMSO stock solutions without preincubation. Cell swelling and osmotic lysis at 37 °C were then tracked by measuring 700 nm light transmittance (DU640 or DU800 spectrophotometer with Peltier temperature control, Beckman Coulter, Brea, CA, USA). Effects of *clag* paralog knockout, protease treatment, or inhibitors were then calculated from the time required to reach a fractional lysis threshold. Inhibitor dose-response data were fitted to a single Langmuir isotherm according to *P* = *a*/[1 + (*x*/*K*_0.5_)] where *P* and *K*_0.5_ represent the normalized solute permeability and the inhibitor affinity constant, respectively.

### 2.8. Computational Analysis and Statistics

Sequence alignments for CLAG paralogs from parasite clones were performed using MultAlin [30]. Posterior probability plots for transmembrane (TM) domain prediction for each CLAG paralog are based on the Phobius algorithm [31].

Numerical data are presented as mean ± S.E.M. from three or more independent trials. Statistical significance was calculated using unpaired Student’s *t*-test or one-way ANOVA with Dunnett’s multiple comparisons test to correct for family-wise errors [32].

## 3. Results

### 3.1. All CLAG Paralogs Traffic via Schizont Rhoptries to the Host Membrane after Reinvasion

As other CLAG paralogs have not been as well characterized as CLAG3, we began our studies by examining the localization and trafficking of these paralogs through epitope tagging each family member in the *C3-TetR* parasite line [29]. *C3-TetR* is an engineered line with several desirable features (Figure 1A). It was produced in the wild-type KC5 line, which carries a single *clag3h* gene; most laboratory lines carry two *clag3* genes that exhibit monoallelic expression through epigenetic switching [14,16,22]. The use of KC5 avoids epigenetic switching. Transfection to produce *C3-TetR* also replaces the last exon of *clag3h* with the corresponding region of Dd2 *clag3.1*; this substitution confers sensitivity to ISPA-28, an inhibitor that is highly selective for channels associated with expression of the Dd2 *clag3.1* last exon. The encoded protein also carries a C-terminal HA epitope tag and a 10× aptamer sequence in the 3′ UTR to recruit the TetR-DOZI fusion protein, permitting conditional knockdown through anhydrotetracyline-dependent expression [33].

Using *C3-TetR*, we performed CRISPR gene-editing to introduce the FLAG epitope tag at the C-terminus of each other *clag* gene product. After transfection, we confirmed integration and used limiting dilution to obtain clonal integrants *C2f_C3-TetR_*, *C8f_C3-TetR_*, and *C9f_C3-TetR_*, where CLAG2, CLAG8, and CLAG9 carry a C-terminal FLAG epitope tag, respectively (Appendix A, Appendix A). We first used these tandem-tagged parasites to examine the stage-specific localization of each paralog. At the schizont stage, where each paralog is maximally transcribed [16,34,35], we found that each CLAG protein co-localized with CLAG3, indicating trafficking of each paralog to rhoptries (Figure 1B, left panels). Upon merozoite egress and invasion of new erythrocytes, each CLAG protein was found in host cytosol, at the erythrocyte surface, and associated with the intracellular parasite (Figure 1B, right panels). The wild-type KC5 parent transfected to carry only the pUF1-Cas9 plasmid did not react with these antibodies, confirming specific recognition of each CLAG member (Figure 1B, WT).

### 3.2. Two-State Behavior and Exposure at the Host Membrane

One of the unusual features of CLAG3 is that it is manufactured in the preceding erythrocyte cycle, transferred to the host cell upon merozoite invasion, and trafficked through multiple parasite compartments as a soluble protein before it undergoes membrane insertion to determine channel-mediated nutrient uptake at the infected erythrocyte’s host membrane [36]. Most such two-state proteins are small and insert into membranes through the formation of β-barrel structures [37], contrasting with CLAG3′s large size and α-helical structure. Release from infected cell membranes through alkaline extraction or freeze-thaw provides strong evidence for CLAG3′s two-state behavior [22,38]. We, therefore, used our FLAG-tagged lines to determine whether each other CLAG paralog also exists in soluble and integral forms. Simple hypotonic lysis released little or no CLAG3 or CLAG9 but released modest amounts of CLAG2 and CLAG8 into the soluble fraction (Figure 2A, first lane in each panel). Alkaline extraction with Na_2_CO_3_ revealed that each paralog is significantly released into the carbonate supernatant (third lanes) and that approximately half of the cellular pool in trophozoite-stage infected cells is integral to membranes (fifth lanes). Thus, like CLAG3, each paralog appears to exist in at least two states within infected cells: as a soluble protein peripherally associated with membranes and as an integral membrane protein.

While early studies established that CLAG3 is exported into the host erythrocyte and is concentrated at the host membrane [35], susceptibility to extracellular protease treatment was required to establish that CLAG3 is integral and exposed at the host membrane surface [22,23]. We confirmed this observation in the *C3-TetR* line (Figure 2A, last lane of the first panel). The cleavage product size, ~40 kDa in *C3-TetR*, indicates cleavage at a site this distance from the protein’s C-terminus. Importantly, this position corresponds to the primary variable region that exhibits sequence and length polymorphisms between geographically divergent parasite isolates (Figure 2B, gray highlight showing the hypervariable region, HVR). Because this cleavage product partitions exclusively into the membrane fraction after Na_2_CO_3_ (Figure 2A, last lane), there are one or more transmembrane domains distal to the HVR. CLAG2 exhibited a similar cleavage product that is also integral to membranes (Figure 2A), implicating a similar protease-mediated cleavage within the smaller CLAG2 HVR (Figure 2B, second alignment graphic). CLAG8 and CLAG9 were not susceptible to extracellular protease, consistent with much less variation between sequences at the presumed surface-exposed site (Figure 2A,B). Computational analysis suggests that each paralog has hydrophobic regions on both sides of this site (Figure 2C, red dash), implicating transmembrane domains on either side of a small surface-exposed region. These observations suggest that each paralog inserts at the host membrane with a similar topology. CLAG3 and CLAG2 appear to have relatively larger loops exposed to plasma, while CLAG8 and CLAG9 have smaller loops that are resistant to extracellular protease treatment (Figure 2D).

### 3.3. Preserved Growth in Nutrient-Restricted Medium and Compensatory Changes in Knockout Lines

Although multiple linkage analysis studies have determined that it is the primary genetic determinant of PSAC-mediated nutrient acquisition by bloodstream *P. falciparum* parasites [15,22,23], a viable CLAG3 knockout was recently produced and characterized (*C3hKO*) [29]. In contrast, RhopH2 and RhopH3, two unrelated proteins that also contribute to PSAC activity, could not be disrupted despite repeated attempts with multiple CRISPR sgRNAs [36]. Although this might reflect a less critical role of CLAG3 than RhopH2 and RhopH3 in PSAC formation, an alternative explanation is that other CLAG paralogs may adequately fulfill CLAG3′s role in the *C3hKO* line. Consistent with this, *rhoph2* and *rhoph3* are single-copy genes without paralogs in *Plasmodium* spp. [36,39,40]. To examine these possibilities, we next produced and cloned viable knockouts of each other *clag* gene in the KC5 wild-type line (Appendix A, Appendix A).

A striking property of the *C3hKO* CLAG3 knockout is that, although it grows at normal rates in the nutrient-rich RPMI 1640-based culture medium typically used for *P. falciparum* culture, it cannot be propagated in PGIM, a nearly identical medium with reduced and more physiological concentrations of the three key nutrients acquired via PSAC (Figure 3A, red bars; [15,29]). We, therefore, examined the growth kinetics of each new CLAG paralog knockout line (Figure 3A). As observed with other wild-type laboratory strains [15], KC5 expands at slower rates in PGIM than in the standard RPMI-based medium because of the reduction in key nutrients to more physiological levels. We confirmed that *C3hKO* expansion is uncompromised in an RPMI-based medium and that its growth is curtailed in PGIM (Figure 3A, red bars). In contrast to *C3hKO*, the *C2KO*, *C8KO*, and *C9KO* lines expanded at rates indistinguishable from the wild-type parent in both RPMI and PGIM, indicating that nutrient uptake is not significantly compromised in these other *clag* knockout lines.

As growth in these knockout lines may be associated with compensatory changes, we next quantified the CLAG3 protein in synchronized trophozoite-stage cultures using immunoblotting. A mouse anti-CLAG3 antibody raised against a C-terminal fragment did not recognize any proteins in the *C3hKO* line, indicating that this antibody is specific and does not recognize other paralogs (Figure 3B). CLAG3 abundance was noticeably increased in the *C2KO* and *C8KO* lines but unchanged in *C9KO*. Band intensity measurements after normalization using an aldolase loading control confirmed this increase (Figure 3B, bottom blot), but these changes did not reach statistical significance in our hands (Figure 3C, *p* = 0.09, one-way ANOVA using *n* = 3 independent trials). We also evaluated changes in transcript levels for each *clag* paralog, as well as *rhoph2* and *rhoph3*, with quantitative RT-PCR (Figure 3D). Here, the *C2KO* and *C8KO* lines revealed significant increases in the transcription of *clag3h*, *clag9*, *rhoph2*, and *rhoph3* (*p* < 0.005, one-way ANOVA, four independent trials for each gene). The *clag2* transcription was also increased in *C8KO.* Notably, the *clag9* knockout clone, *C9KO*, did not exhibit increased transcription of any of these genes, paralleling insignificant changes observed for *C3hKO* in a recent study [29]. Thus, increased transcription of PSAC-associated genes in the *C2KO* and *C8KO* suggests compensatory changes that maintain adequate PSAC-mediated nutrient uptake to allow growth in the nutrient-restricted PGIM; these transcriptional changes could also facilitate parasite survival through unrelated mechanisms.

### 3.4. PSAC Transport Phenotypes of CLAG Knockout Lines

*C3hKO* exhibits reduced permeability to organic solutes, with 60–75% lower uptake rates for sorbitol, proline, and the organic cation phenyl-trimethylammonium (PhTMA^+^) [29] (Figure 4A,B, red bars). Because the uptake of these divergent solutes occurs almost exclusively via PSAC in infected cells, it is remarkable that *C3hKO* incurs only an incomplete reduction in these permeabilities. Notably, conditional knockdown of RhopH2 or RhopH3 quantitatively abolishes PSAC-mediated transport of organic and inorganic solutes [36]. We, therefore, measured solute permeabilities in the new paralog knockout lines using continuous tracking of osmotic lysis in sorbitol, PhTMA^+^, and proline solutions [29]. In contrast to *C3hKO*, the *C2KO*, *C8KO*, and *C9KO* lines did not exhibit reduced permeability to these solutes. Instead, *C2KO* and *C8KO* had modestly faster osmotic lysis in each solute (Figure 4A,B), which were statistically significant in experiments that measured PhTMA^+^ uptake (*p* < 0.02, one-way ANOVA, 3–4 independent trials for each knockout clone).

As extracellular protease treatment cleaves CLAG3 and CLAG2 within their hypervariable domains (Figure 2A), we then examined the effects of this treatment on PSAC-mediated transport in these knockout lines. Although many proteins on the infected erythrocyte surface are presumably cleaved, linkage analysis and DNA transfection studies have determined that CLAG3 proteolysis fully accounts for reduced solute transport after this treatment [23]. Consistent with prior studies using other lines, we found that sorbitol uptake into KC5 wild-type was reduced by ~90% upon pronase E treatment (Figure 4C,D). Remarkably, *C3hKO*’s lower sorbitol permeability was not measurably compromised by this treatment, but uptake into *C2KO*-infected cells retained their susceptibility to protease treatment. This result suggests that reduced sorbitol transport after proteolysis results primarily from the steric hindrance of the pore by a cleaved CLAG3 extracellular HVR. This steric hindrance model is supported by previously reported DNA transfection experiments and strain-specific differences in susceptibility to proteases with distinct cleavage profiles [23], but other mechanisms remain possible in the absence of an atomic-resolution ion-channel structure. As *C2KO*-infected cells have an unchanged protease susceptibility, this experiment could not support a direct contribution of CLAG2 to the formation of the channel pore.

### 3.5. No Significant Change in PSAC Pharmacology

The *C3hKO* line also exhibits marked changes in inhibitor affinity with reduced efficacy of phloridzin, ISG21, and ISPA-1 block when compared to its KC5 wild-type parent [29]. These reduced affinities suggest that the binding sites for these inhibitors are partially defined by CLAG3. We hypothesized that these small molecule inhibitors interact with a specific pocket on the channel lined by one or more CLAG3 residues that are distinct from those on CLAG paralogs that take over channel formation upon *clag3* knockout. We, therefore, examined inhibitor efficacy in each of the *clag* knockout lines and used the well-known PSAC inhibitors furosemide and phloridzin [41], as well as ISG21, a highly potent inhibitor identified through high-throughput screens [15]. This broad-spectrum inhibitor blocked KC5 channels with high affinity (*K*_0.5_ of 2.8 ± 0.5 nM) but was ~5-fold less effective against *C3hKO* (Figure 5A,B). Phloridzin, but not furosemide, also exhibited compromised affinity against *Ch3KO* channels (Figure 5B). In contrast to *Ch3KO*, channels on the *C2KO*, *C8KO*, and *C9KO* knockout lines did not incur changes in affinity with any of these inhibitors. These findings support the hypothesis that specific residues on CLAG3, which differ from the corresponding sites on the other CLAG paralogs, improve affinities for these particular inhibitors. Rather than reflecting a greater blocking capacity of CLAG3-associated channels, we suspect that higher expression of CLAG3, compared to other paralogs (Figure 3D), led to the identification of CLAG3-favoring blockers in low- and high-throughput searches. Presumably, inhibitor screens using the *C3hKO* line would find inhibitors that preferentially block channels associated with each other paralog.

## 4. Discussion

Erythrocytes infected with malaria parasites have increased permeability to diverse solutes, including anions, organic cations, and nutrients such as sugars, amino acids, and purines [42,43,44,45]. These increases were originally identified in ex vivo experiments from malarious monkeys [46] and were primarily studied with macroscopic osmotic fragility and tracer flux methods until the turn of the century. Various mechanisms, including one or more transporters, membrane leaks, a debated parasitophorous duct, and endocytosis, were proposed based on these studies [47,48,49]. The patch clamp of *P. falciparum*-infected erythrocytes resolved these mechanistic uncertainties by identifying the plasmodial surface anion channel (PSAC) [21]. Subsequent patch-clamp studies suggested a number of distinct channels, which were proposed to be host-derived channels upregulated by the intracellular parasite [50,51,52]. The in vitro selection of transport mutants then revealed altered PSAC gating and reduced uptake of organic solutes, suggesting a role for parasite genetic elements. These controversies were finally resolved through agnostic genetic mapping studies using ISPA-28, a strain-specific PSAC inhibitor identified in high-throughput screens [22]. When combined with DNA transfection, these genetic mapping studies implicated CLAG3. Protease susceptibility studies revealed that CLAG3 is integral to the host membrane, suggesting a direct contribution to the nutrient uptake channel. Years before the identification of its role in transport, biochemical studies had established that CLAG3 associated with RhopH2 and RhopH3, two conserved and unrelated parasite proteins [53]. Subsequent molecular and cellular studies have revealed that these proteins also contribute to PSAC-mediated nutrient and ion transport [36,39], but the precise roles of each subunit have remained unclear.

While the link between PSAC activity and CLAG3 has been intensively studied [22,54], possible contributions from three paralogs encoded by *clag* genes on *P. falciparum* chromosomes 2, 8, and 9 (termed CLAG2, CLAG8, and CLAG9, respectively) have not been examined to date. Here, we used DNA transfection, transcript, and protein studies along with transport measurements to examine the potential roles of these paralogs. Epitope tagging of each paralog revealed that each paralog is packaged in merozoite rhoptries, transferred to the next erythrocyte, and trafficked to the host membrane, as previously established for CLAG3. Also, like CLAG3, each paralog exists in at least two pools within infected erythrocytes: an alkaline-extractable form weakly associated with membranes and a form that is integral to membranes. Each member exhibits the greatest variation amongst sequenced clones at a region approximately 30–40 kDa from its C-terminus, with the larger variant regions of CLAG3 and CLAG2 exposed to plasma upon insertion in the host erythrocyte membrane; CLAG8 and CLAG9 exhibit much less variation at this site and do not exhibit protease susceptibility, suggesting minimal or no exposure to plasma. Conserved hydrophobicity on either side of this variant region suggests that CLAG3 and its paralogs have similar transmembrane topologies at the host membrane.

We used CRISPR/Cas9 transfection to disrupt each paralog and found that individual *clag2* or *clag8* knockouts exhibited modest increases in CLAG3 abundance. RT-PCR extended this finding by revealing statistically significant upregulation of *clag3h*, *clag9*, *rhoph2*, and *rhoph3* in our CLAG2 and CLAG8 knockout lines. The *clag2* transcription was also significantly increased in the CLAG8 knockout; *clag8* increased in the CLAG2 knockout but did not reach significance. These transcriptional changes establish that the knockout of each paralog leads to compensatory changes in other genes linked to PSAC formation, supporting overlapping roles for these proteins.

These molecular changes were associated with a significantly increased permeability to PhTMA^+^, an organic cation whose accumulation occurs almost exclusively via PSAC [44]. This increase might reflect an increase in the PSAC copy number that results from the increase in CLAG3 abundance. Alternatively, it could reflect changes in PSAC selectivity for permeant solutes due to the loss of individual CLAG paralogs that function as subunits of the channel [29]. In either case, these findings represent early, though inconclusive, experimental evidence supporting the roles of CLAG2 and CLAG8 in PSAC formation. Unrelated roles at the host membrane remain possible and should be explored further. We propose an expansion of the *clag* gene family to allow redundancy in the formation of the essential nutrient uptake channel; this model suggests partially overlapping contributions through the formation of a channel complex consisting of CLAG, RhopH2, and RhopH3 subunits.

In addition to the expansion of the *clag* gene family in *Plasmodium* spp., diversity and redundancy are also promoted by transcriptional control through monoallelic expression and switching between the two *clag3* paralogs in most lines and variable *clag2* expression in *P. falciparum* [16,55]. The benefits to the parasite are not well-understood but may include fine-tuning of channel-mediated nutrient uptake, as evidenced by switching from preferential expression of *clag3.2* in human infections to expression of *clag3.1* upon adaptation to in vitro culture [25,56]. Switching may also promote immune evasion by the silencing of CLAG paralogs when an antibody response to an exposed epitope is generated. Epigenetic regulation has also been shown to permit escape from toxicity due to toxin uptake, as silencing of *clag3* and/or *clag2* is the primary mechanism of acquired resistance to blasticidin S, which enters infected cells primarily via PSAC [24,25].

Systematic knockouts of individual CLAG paralogs, as we performed, revealed several important differences. Most importantly, while each paralog is individually dispensable under standard in vitro culture conditions using nutrient-rich RPMI 1640 medium, loss of CLAG3 in the *C3hKO* parasite aborts expansion in PGIM, a modified medium with more physiological levels of isoleucine, glutamine, and hypoxanthine [15]; isoleucine appears to be especially critical as it cannot be obtained through hemoglobin digestion [57,58]. In marked contrast, *C2KO*, *C8KO,* and *C9KO* all exhibited unabated growth in PGIM (Figure 3A). One explanation for this preserved growth is that the higher levels of CLAG3 in most parasite clones may be quantitatively sufficient to permit sustained channel formation and nutrient uptake (Figure 3). Another possibility is that channels formed with distinct paralogs permit preferential transport of certain nutrients and that the nutrients with altered levels in PGIM are primarily transported via CLAG3-associated channels. This second possibility parallels a proposed difference in nutrient acquisition by CLAG3.1 and CLAG3.2, the two isoforms present in most parasite clones [56]. The preserved growth of *C2KO*, *C8KO,* and *C9KO* in PGIM could also reflect the lesser or negligible roles of these paralogs in channel-mediated nutrient uptake.

Another notable difference between the knockout lines is that, while CLAG3 and CLAG2 both carry extracellular loops susceptible to protease treatment (Figure 2), pronase treatment significantly compromises channel-mediated uptake in wild-type and *C2KO* lines but has no clear effect on uptake by *C3hKO*-infected cells (Figure 4). This finding is consistent with a proposal that the channel pore is sterically occluded by the cut ends of the CLAG3 HVR produced by pronase treatment [23]. In this proposed model, loss of CLAG3 in *C3hKO* abolishes this steric hindrance and renders the uptake protease-insensitive. If this model is eventually proven correct, our data would suggest that the shorter cut ends generated by cleavage of the smaller CLAG2 HVR may not be long enough to occlude the channel pore; it is also possible that CLAG2 does not directly contribute to the channel pore.

Our studies also reveal that, while *C3hKO* exhibits marked changes in PSAC pharmacology, disruption of each other paralog has negligible effects on the block by ISG21, phloridzin, and furosemide (Figure 5). ISG21 is particularly important because this compound is the parent compound for a recently reported hit-to-lead drug discovery project targeting PSAC [59]. This finding may also reflect a higher expression of CLAG3 in wild-type clones and preferential identification of inhibitors that block sites on associated channels. In this scenario, inhibitor screens using the *C3hKO* line may reveal inhibitors that specifically interact with channels formed using CLAG2, CLAG8, and/or CLAG9. Such inhibitors could convincingly implicate these paralogs in the direct formation of PSAC. They would also provide important structure-function insights into the channel pore and may be starting points for therapies targeting parasite nutrient acquisition.

## 5. Conclusions

The *clag* multigene family is strictly conserved in all examined malaria parasite species, but absent from other genera. While one encoded protein, CLAG3, has been extensively studied and linked to a nutrient uptake channel at the host erythrocyte membrane of infected cells, proteins encoded by other *clag* genes have not been as well studied. Here, we used DNA transfection to engineer epitope-tagged versions and knockout lines for each other member in *P. falciparum*, the most virulent agent of human malaria. Our studies reveal that each encoded paralog exhibits trafficking identical to CLAG3. Each also exists in both carbonate-extractable and integral forms within infected erythrocytes. The integral forms embed within host erythrocyte membranes with similar topologies, though protease susceptibility experiments suggest key differences. Compensatory changes in CLAG3 abundance parallel transcriptional changes in the CLAG2 and CLAG8 knockout lines, which also exhibit increased permeability to a reporter solute with high permeability through the nutrient channel. These findings provide important insights into the proteins encoded by an unusual gene family conserved in malaria parasites.

## Figures and Tables

**Figure 1 microorganisms-12-01172-f001:**
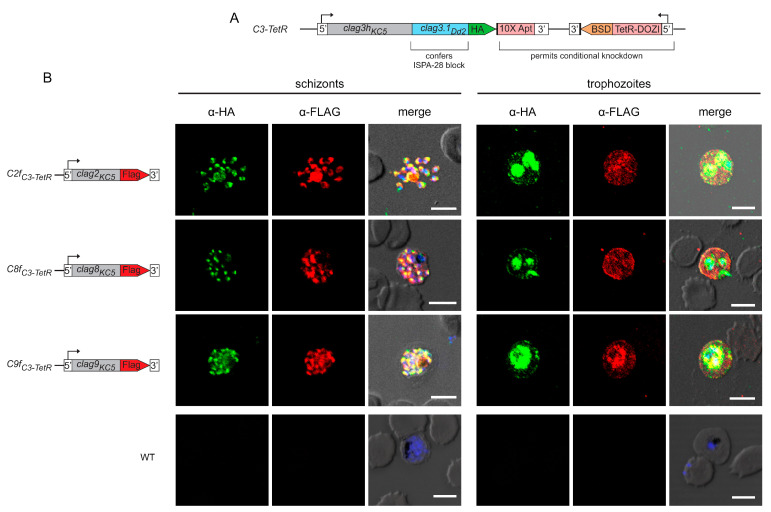
Paralogs exhibit stage-specific localization and trafficking similar to CLAG3. (**A**) Schematic showing the engineered *C3-TetR* line. This parasite carries a single chimeric *clag3h* product with a 1xHA epitope tag and conditional knockdown through TetR-DOZI interaction with a 10× aptamer sequence in the *clag3h* 3′UTR. (**B**) Indirect immunofluorescence antibody (IFA) images for double-tagged parasites. While anti-HA recognizes CLAG3, anti-FLAG recognizes a separate CLAG protein in each row, as indicated with the schematic to the left. Note that each paralog co-localizes with CLAG3 in rhoptries at the schizont stage (puncta, **left** panels) and that each is exported to the host membrane at the trophozoite stage (**right**). The parental control is not recognized by either antibody (**bottom row**). Scale bars, 5 µm.

**Figure 2 microorganisms-12-01172-f002:**
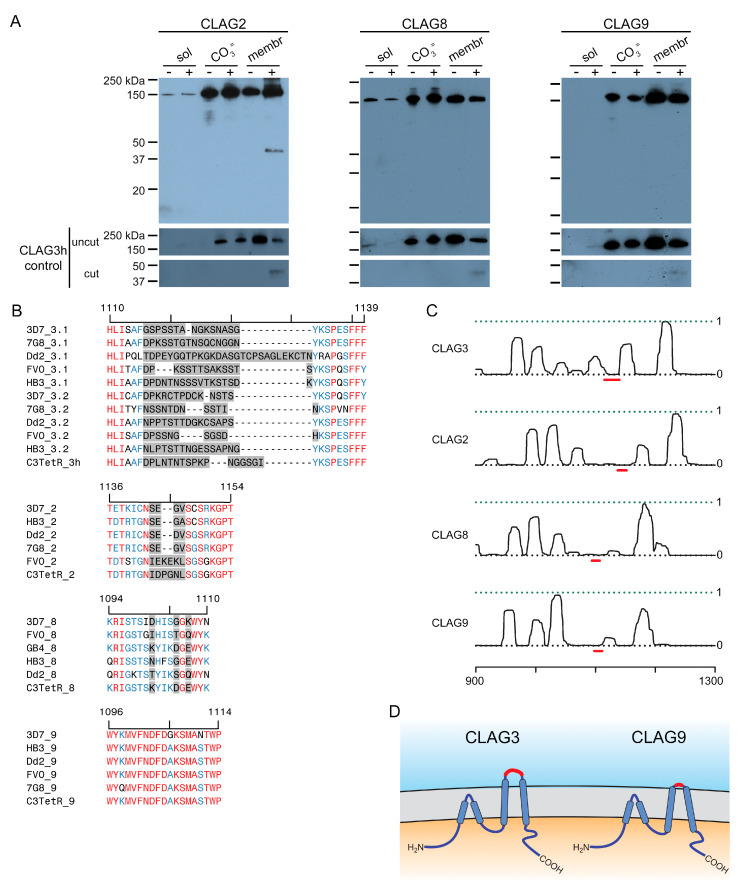
Two-state behavior and similar topologies at the host membrane. (**A**) Immunoblots using the anti-FLAG antibody, showing that indicated CLAG paralogs are minimally released upon hypotonic lysis (sol). Alkaline treatment reveals both extractable and integral membrane fractions (CO_3_^=^ and membrane, respectively). Samples with and without pronase E treatment of intact cells are shown to determine each paralog’s susceptibility to extracellular protease treatment. Each sample was probed with anti-CLAG3 antibody (**bottom**) to validate the fractionation protocol. (**B**) Alignment of the variant regions from top to bottom: CLAG3.1 and CLAG3.2, CLAG2, CLAG8, and CLAG9. Each group contains divergent *P. falciparum* lines and the *C3-TetR* line used in these studies. The ruler above each group reflects the residue number in the reference 3D7 clone. In each group, highly variant residues are highlighted in gray, while identical and conserved residues are in red and blue, respectively. Note that CLAG3 has the largest HVR and that other paralogs have distinct degrees of variation at this site. (**C**) Posterior probability plots for transmembrane (TM) domain prediction for each paralog (solid black line) over a region that includes the variant region from panel (**B**) (red line). Note the high probability of a TM distal to each variant region, consistent with membrane partitioning of the protease-induced cleavage product in CLAG3 and CLAG2. Each protein also likely has one or more transmembrane domains proximal to the variant region. The ruler at the bottom indicates the residue position from the unprocessed N-terminus of each paralog. (**D**) Schematic showing a model of similar topologies at the host membrane. Variant, surface exposed sites are shown in red. Notice the smaller, less exposed surface loop on CLAG9.

**Figure 3 microorganisms-12-01172-f003:**
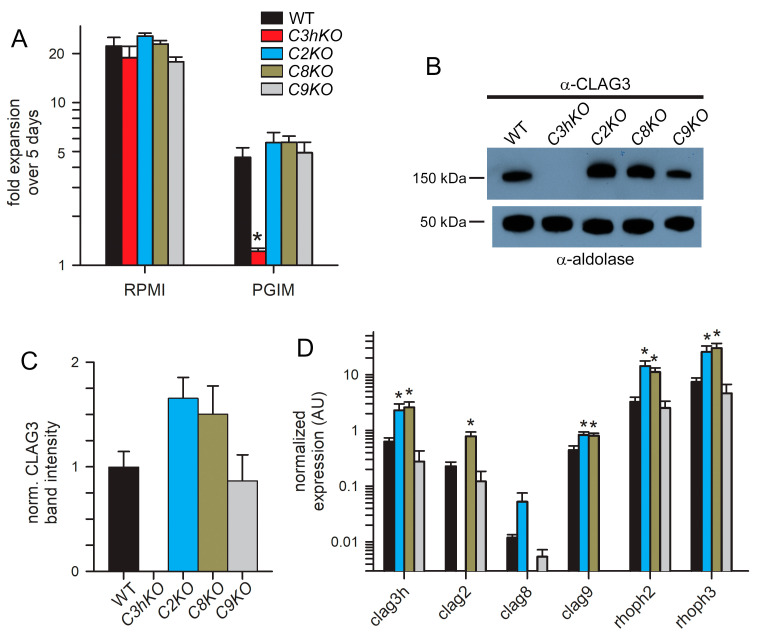
Growth rates and compensatory changes in each CLAG knockout line. (**A**) Mean ± S.E.M. expansion of wild-type and CLAG knockout cultures over 5 days in indicated media. Black bars, wild-type KC5; red, *C3hKO*; blue, *C2KO*; dark yellow, *C8KO*; gray, *C9KO*. *n* = 3 independent trials each. (**B**) Immunoblots using total trophozoite-infected cell lysates from indicated parasites probed with anti-CLAG3 (**top**) and anti-aldolase as a loading control (**bottom**). (**C**) Mean ± S.E.M. ImageJ quantification of band intensities from immunoblots as in panel B after correction for the aldolase loading control band intensity and normalization of WT to 1.0. *n* = 3 trials each. (**D**) Mean ± S.E.M. normalized expression of indicated *clag* and *rhoph* genes in the WT (black) and CLAG knockout lines (blue, *C2KO*; dark yellow, *C8KO*; gray, *C9KO*), calculated according to 2^−ΔCt^ using PF07_0073 as an internal loading control; *n =* 4 trials each. For all panels, asterisks indicate *p <* 0.05, one-way ANOVA with Dunnett’s multiple comparisons test.

**Figure 4 microorganisms-12-01172-f004:**
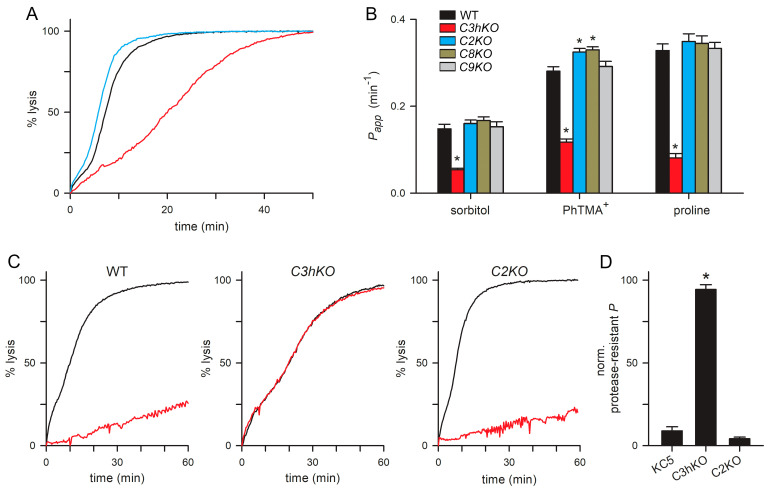
PSAC-mediated transport in each CLAG knockout line. (**A**) Sorbitol-induced osmotic lysis kinetics for wild-type, C3hKO, and C2KO (black, red, and blue traces, respectively). Note, the slower kinetic and increased time to 50% lysis for C3hKO but not for C2KO relative to the wild-type parent. (**B**) Mean ± S.E.M. permeabilities of indicated solutes, calculated as the reciprocal of the time to 50% lysis from osmotic lysis kinetics experiments. Black bars, wild-type KC5; red, C3hKO; blue, C2KO; dark yellow, C8KO; gray, C9KO. *n* = 3–8 independent trials for sorbitol and PhTMA^+^; 2–5 trials for proline. Asterisks, *p* < 0.05. (**C**) Sorbitol-induced lysis kinetics for indicated clones without and with pronase E pretreatment (black and red traces, respectively). Note that C3hKO’s reduced sorbitol permeability is not susceptible to proteolytic treatment, in contrast to the wild-type and C2KO. (**D**) Mean ± S.E.M. reduction in permeability upon pronase E treatment, normalized to 100% for no effect. Asterisk, *p* < 0.0001.

**Figure 5 microorganisms-12-01172-f005:**
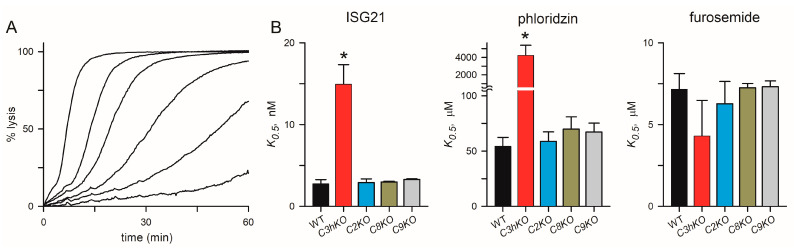
Channel pharmacology in CLAG knockout lines. (**A**) Sorbitol-induced lysis kinetics for the KC5 wild-type line with 0, 2.5, 5, 10, 20, and 50 nM ISG21 (left-to-right traces). This potent inhibitor slows sorbitol uptake in a dose-dependent manner. (**B**) Inhibitor *K*_0.5_ values for indicated parasite clone for ISG21, phloridzin, and furosemide (**left-to-right** panels). Asterisks indicate *p <* 0.0001, one-way ANOVA with Dunnett’s multiple comparisons test.

## Data Availability

The original contributions presented in this study are included in the article and Appendix A, further inquiries can be directed to the corresponding author.

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
