# Peer review of "Plasmodium falciparum CLAG Paralogs All Traffic to the Host Membrane but Knockouts Have Distinct Phenotypes"

_microorganisms, 2024, doi:10.3390/microorganisms12061172_

Round 1
Reviewer 1 Report
Comments and Suggestions for Authors
This manuscript reports the characterization of all paralogs of the Plasmodium falciparum CLAG family. While CLAG3s have been previously characterized and play a fundamental role in the formation of the PSAC channel (involved in solute transport), CLAG2, CLAG8 and CLAG9 had not been previously characterized in detail. The authors generated new transgenic parasite lines with epitope-tagged CLAG proteins or KO of the different CLAG genes. After characterizing these transgenic lines using IFA, cell fractionation, transcriptional analysis, transport studies (using osmotic lysis as a surrogate), protease sensitivity and pharmacological sensitivity, the authors conclude that CLAG paralogs beyond CLAG3 contribute to the formation of PSAC and therefore to solute transport.
The results are well presented and the manuscript is well written. The experiments are performed rigorously and the quality of the data is in general excellent. However, I have concerns with the design and interpretation of two specific experiments, as detailed below. Furthermore, in my opinion, the main conclusion of the manuscript is not well supported by the data presented. While it is possible that CLAG2, CLAG8 and CLAG9 are involved in PSAC formation, other scenarios are possible and cannot be excluded. In fact, in the light of the data presented, it appears very likely that these other CLAGs are not involved in the transport of the solutes tested. If the authors are willing to reconsider their interpretation of the results and modify the conclusions, this manuscript could be an important contribution to our understanding of the CLAG family.
Major comments.
1. Controls are missing for the experiments presented in Fig. 2A. The same samples (or even the same membranes) should be tested with antibodies against control proteins know to be specific for the different cellular fractions. Without this important control, it is impossible to know if all CLAGs actually occur as both peripheral and integral membrane proteins, or the fractions are not sufficiently pure.
2. The increased expression of all RhopH complex genes (including clag genes and also rhoph2 and rhoph3) in some KO lines, reported in Fig. 3, can be explained by differences in the age of the parasites between the samples collected from the different lines. Differences in the age of the parasites of only a few hours (e.g., 3 h) can result in large differences in the expression of these genes, with a similar or even larger magnitude as the differences reported here between WT and transgenic lines. This is a common artifact in malaria research. It is impossible to adjust the average parasite age of different cultures precisely (e.g., less than 3 h differences between cultures) with the approach used here (sorbitol synchronization only). Furthermore, age differences of only a few hours cannot be detected by microscopic examination of Giemsa-stained smears (the validation used here to confirm that different lines were at the same stage, according to the Methods).
In my opinion, this caveat invalidates the results of Fig. 3. To avoid repeating the experiments with cultures synchronized to a defined narrow age window, one possibility for the authors would be to simply test (in the same samples) the expression of unrelated genes that show the same temporal pattern of expression as rhoph genes (the appropriate control genes could be identified from PlasmoDB). If these control genes also show increased expression in the same KO lines, this would clearly indicate that the apparent increased expression of rhoph genes can be explained by differences in the age of the cultures from which RNA samples were collected, rather than by a specific compensatory transcriptional response.
3. The only result that clearly supports a role for CLAG2 and CLAG8 in PSAC formation is the reported compensatory increase in the expression of other rhoph complex genes in the KO lines. However, as discussed in my previous comment, the results of these experiments need to be reassessed.
Other results in the manuscript do not support a role for CLAG2, 8 or 9 in the transport of the solutes tested (those at low levels in PGIM and those used for osmotic lysis assays), and rather indicate that CLAG3 is the only CLAG protein that accounts for most, if not all, of the transport activity for these solutes:
-Fig. 3A shows similar growth for WT or CLAG2, CLAG8 and CLAG9 KO lines, both in RPMI and PGIM. Only CLAG3 mutants show a markedly different profile.
-Osmotic lysis experiments in Fig. 4 show almost identical profiles for WT or CLAG2, CLAG8 and CLAG9 KO lines, and very different profiles only for CLAG3 mutants. The differences for PhTMA permeability (not for other compounds) in CLAG2 and 8 mutants are statistically significant but of very low magnitude, of unclear biological relevance. Experiments in Fig. 4C neither provide support to the idea that CLAG2 contributes to PSAC activity.
-The results of the experiments with channel inhibitors (Fig. 5) also show an identical profile for WT or CLAG2, CLAG8 and CLAG9 KO lines. In this case, the result can be explained by the specificity of the inhibitors, but in any case, it does not provide support to the idea that all CLAGs participate in PSAC formation.
The main conclusion of the article should be revised, with changes in the title, abstract, results and discussion.
Minor comments.
-Line 54 and elsewhere. Only clag2 and clag3 genes, but not clag8 and clag9, have been clearly shown to have variable expression. This fits with the observation that only CLAG2 and CLAG3s seem to have a large region exposed at the surface, according with the protease sensitivity experiments in Fig. 2A.
-Fig. 1B, trophozoites. Surface localization is apparent for CLAG8-FLAG, but not for the other proteins. This should be reflected in the text.
-Fig. 2B. How was the position of the HVR determined for CLAG8 and CLAG9? Especially for CLAG9, it appears as there is no HVR. Additionally, the schematic and the text (“flank”, line 312) suggest that hydrophobic regions are immediately adjacent to the HVR, but this only seems to be the case for CLAG3. Please phrase in a way that describes this more accurately. Please also revise the statement in the Discussion (line 474).
-In fig. 3A (and others), a legend with the color for each parasite line within the figure would be helpful.
-Line 414 and Discussion. While steric hindrance of the pore by a cleaved CLAG3 and not by a cleaved CLAG2 provides a possible explanation, a more straight forward interpretation of these results is that CLAG3 cleavage may render PSAC non-functional, whereas CLAG2 may not participate in PSAC formation and therefore transport would be unaffected by CLAG2 cleavage. This possibility should also be discussed
-Fig. S1 and S2. Some of the gel images are cropped just at the middle of the position of some bands.
Reviewer 2 Report
Comments and Suggestions for Authors
The manuscripts is done well.
However, I believe that the inclusion of a clear subheading for "Conclusion and Recommendation@ might be appropriate for the sake of the general readers.
Reviewer 3 Report
Comments and Suggestions for Authors
1.This manuscript has some reference value in plasmodium biology, but it has some shortcomings.
2.In the knockout of C2fC3-TetR, C8fC3-TetR, and C9fC3-TetR, some control reference gene or reverse proof reference gene tests should be added.
3.There are some English words that are not written properly.
4.Check experimental data and graphs.
Comments on the Quality of English Language/
Round 2
Reviewer 3 Report
Comments and Suggestions for Authors
The author has made a lot of revisions, and it is recommended to publish them after modification
Comments on the Quality of English Language/
Author Response
We have completed the revision and believe the changes have improved presentation.